# Manipulating single excess electrons in monolayer transition metal dihalide

Min Cai[1], Mao-Peng Miao[1], Yunfan Liang[2], Zeyu Jiang[2], Zhen-Yu Liu[1], Wen-Hao Zhang [1,3], Xin Liao[1], Lan-Fang Zhu[1], Damien West[2], Shengbai Zhang[2] & Ying-Shuang Fu [1,3,4] ✉

Polarons are entities of excess electrons dressed with local response of lattices, whose atomic-scale characterization is essential for understanding the many body physics arising from the electron-lattice entanglement, yet difficult to achieve. Here, using scanning tunneling microscopy and spectroscopy (STM/STS), we show the visualization and manipulation of single polarons in monolayer $CoCl_2$, that are grown on HOPG substrate via molecular beam epitaxy. Two types of polarons are identified, both inducing upward local band bending, but exhibiting distinct appearances, lattice occupations and polaronic states. First principles calculations unveil origin of polarons that are stabilized by cooperative electron-electron and electron-phonon interactions. Both types of polarons can be created, moved, erased, and moreover interconverted individually by the STM tip, as driven by tip electric field and inelastic electron tunneling effect. This finding identifies the rich category of polarons in $CoCl_2$ and their feasibility of precise control unprecedently, which can be generalized to other transition metal halides.

The interaction of electrons and lattice plays a pivotal role in the many body physics of solids[1,2]. While the lattice vibrations can be treated perturbatively to the electron transport in the weak-coupling limit, the lattice anti-adiabatically responds to the excess electron in the strong coupling limit[3]. In a dielectric lattice, strong electron-lattice coupling results in the displacement of ions adjacent to the excess electron, creating a potential well to get the electron trapped[4,5]. As a result, the excess electron is dressed by local lattice distortions when propagating through polarizable solids, forming a composite quasiparticle, named as polaron[6–9].

Polarons have attracted extensive interests in multidiscipline fields[8,9], because of their key impacts on many physical processes, as exemplified in charge transport[10], colossal magnetoresistance[11], high-temperature superconductivity[12], as well as on chemical reactivity[13,14] and functional materials with thermoelectric and multiferroic behavior[15,16]. Depending on their multi-facet properties, polarons can be detected with various probes, manifesting themselves as large mass

in charge transport[17], associated band dispersion in angle-resolved photo-emission spectra[18–20], absorption and luminescence spectra of their transition between the ground state and excited state[21,22], as well as their spin character in electron spin resonance spectra[23,24], etc. Despite of their existence probed with above ensemble averaged techniques, a direct visualization and spectroscopic characterization of individual polarons at atomic scale are highly desirable for elucidating their properties[25,26]. This is particularly important in material systems where different types of polarons coexist, exhibiting distinct atomistic and electronic structures[27].

Scanning tunneling microscopy (STM) has the capability of atomic resolution imaging and spectroscopic characterization of the local density of states, which is highly suited for the probe of single polarons. However, the dielectric crystals hosting polarons are insulators, posing challenge for STM tunneling measurement. This restricts pioneering STM studies on polarons to systems of narrow gap semiconductors or doped insulators[27–29]. To overcome that

[1]School of Physics and Wuhan National High Magnetic Field Center, Huazhong University of Science and Technology, 430074 Wuhan, China. [2]Department of Physics, Applied Physics and Astronomy, Rensselaer Polytechnic Institute, Troy, NY 12180, USA. [3]Institute for Quantum Science and Engineering, Huazhong University of Science and Technology, 430074 Wuhan, China. [4]Wuhan Institute of Quantum Technology, 430206 Wuhan, China. ✉e-mail: yfu@hust.edu.cn

hurdle, an alternative arena is to study polaron thin films on conductive substrates, which allows electron tunneling through[30]. However, the conductive substrate may enhance charge screening and strongly interacts with the thin films, potentially destabilizing the polarons. Our strategy is to use graphene or highly oriented pyrolytic graphite (HOPG) substrate, which have not only low carrier density but also weak van der Waals (vdW) interaction with the supporting films[31]. The system of choice is monolayer transition metal dihalide, which possesses ionic bonding and strong correlation effects[32].

In this work, we report the visualization and manipulation of single polarons in monolayer $CoCl_2$ on HOPG substrate with STM. Two types of polarons are discovered in $CoCl_2$, which exhibit stunning differences in occupation sites, appearances and polaronic states, etc. Their origins are elucidated with first-principles calculations. More importantly, both types of polarons can be manipulated, with controlled creation, movement, annihilation, and interconversion among different types, demonstrating versatile tunability of this polaron system. The electric field exerted from the tip and the inelastic electron tunneling effect are found to be the driving force for manipulating the polarons. This paves the way of studying polarons at atomic limit.

The experiments were performed with a custom-made cryogenic Unisoku 1500 STM system[33]. Monolayer $CoCl_2$ films are grown by molecular beam epitaxy (MBE) on HOPG substrate. The first principles calculation is carried out with hybrid functional HSE06 approach. The detailed methods are depicted in Supplementary Information.

## Results

### Identification and manipulation of polarons

Bulk $CoCl_2$ is a van der Waals crystal that belongs to trigonal $\bar{R}3m$ space group[34]. Its each vdW layer consists of a triangular lattice of Co layer sandwiched between two Cl layers, with each Co cation octahedrally coordinated by six Cl anions, forming 1-T structure [Fig. 1a]. Figure 1b shows a typical topographic image of monolayer $CoCl_2$ film on HOPG substrate. The apparent height of the monolayer film is measured as 7.8 Å at 2 V and indicate prominent bias dependence [Supplementary Fig. 1], demonstrating its distinct spectroscopic features from the

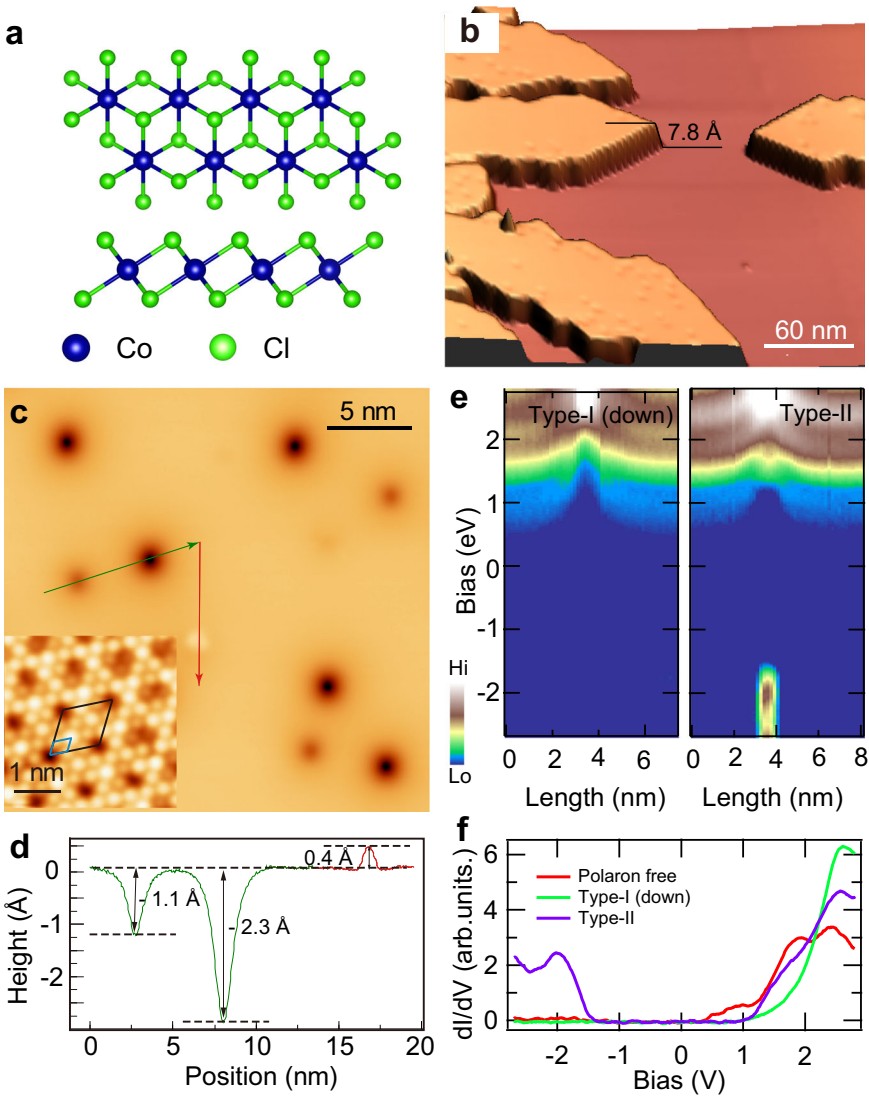

**Fig. 1 | Topography and spectra of polarons. a** Top and side view of the crystal structure of monolayer $CoCl_2$. **b** Large-scale STM topographic image of $CoCl_2$ film ($V_s = 2$ V, $I_t = 5$ pA). **c** STM image of three types of polarons ($V_s = 1.2$ V, $I_t = 10$ pA). The inset shows atomic resolution of $CoCl_2$ ($V_s = 0.3$ V, $I_t = 20$ pA). The blue (black) rhombus marks the unit lattice of the Cl atoms (a 3 × 3 moiré pattern). **d** Line profile of the three kinds of polarons taken along the green and red arrows marked in (**c**). **e** 2D conductance plot across the type-I (down) and type-II polarons. Spectroscopic condition: $V_s = 2.8$ V, $I_t = 150$ pA. **f** Point spectra measured on the type-I (down), type-II polaron center and the polaron-free area of the film.

substrate. Atomic resolution image of $CoCl_2$ imaged at 0.3 V displays a triangular lattice of Cl atoms, whose measured lattice constant 3.5 Å is consistent with the reported value [Fig. 1c, inset][34]. There is a moiré pattern with a $3 \times 3$ periodicity superimposed on the atomic lattice. Multiple moiré patterns are observed in surface domains characterized by different relative angles between the grown $CoCl_2$ layer and the graphene substrate [Supplementary Fig. 2].

When imaging the monolayer $CoCl_2$ at a larger bias of 1.2 V, there appear two kinds of depressed entities coexisting with another kind of protruded entities randomly distributed over the film [Fig. 1c, Supplementary Fig. 3], whose apparent heights are measured as −1.1 Å, −2.3 Å, and 0.4 Å, respectively [Fig. 1d]. As shown later, those three kinds of entities are all polarons, instead of crystal defects. Hereafter, the shallow (deep) entities are named as type-I (down) [type-I (up)] polarons, and the protruded entities are type-II polarons. Their appearances change drastically with imaging bias. Both type-I (down) and type-I (up) polarons enlarge monotonically with decreasing bias from 1.2 V to 0.4 V, and become invisible below 0.3 V [Supplementary Fig. 4]. The type-II polaron appears as a protrusion above 1.2 V and below −0.5 V [Supplementary Fig. 5], but becomes a depression and indistinguishable with the type-I (down) polaron between 0.8 V and 1.0 V [Supplementary Fig. 6]. The population ratio between the type-I (down) and type-II polarons is about 50:1.

The bias-dependent polaron morphologies are related with their electronic properties [Supplementary Note 1], which are subsequently characterized with spectroscopic measurements. The conduction band minimum (CBM) of the bare monolayer $CoCl_2$ locates at 0.25 eV that is spatially uniform [Supplementary Fig. 3], and the valence band maximum should be below the measured spectroscopic range of −3 eV [Fig. 1f]. Upon approaching these polarons, their spectra all bend upward, but with different magnitudes [Fig. 1e and supplementary Fig. 7]. Specifically, the CBM shifts to 0.5 eV (0.7 eV) for the type-I (down) [type-I (up)] polaron [Supplementary Fig. 7]. Compared to the type-I (down) polaron, the type-II polaron induces similar shift of CBM, above which its conductance is larger [Fig. 1e]. While the type-I (down) and type-I (up) polarons have no noticeable polaronic states, the type-II polaron exhibits clear polaronic state peak at −2 eV [Fig. 1e, f].

The upward band bending conforms to the host of excess electrons by polarons, elevating the local chemical potential. It is noted that crystal defects of charge acceptors could also accommodate excess electrons. To discern the two scenarios, we perform tip manipulations to the defect-like entity. Upon the tip is on top of a type-I (down) entity with the tunneling junction set at 3.0 V/7 nA, the tip height becomes unstable [Supplementary Fig. 9], implying state switching of the entity. Subsequent imaging at low bias shows the entity is annihilated [Fig. 2a, b]. This unambiguously proves it is a polaron, instead of a real crystal defect. The type-I (down) polaron can be created by applying a negative bias of − 3.5 eV [Fig. 2a, b]. The created polaron indicates identical features as the naturally formed ones. We can laterally move the type-I (down) polaron, which stably follows the tip trajectory at 3.0 V/3 nA or −3.5 V/0.1 nA [Fig. 2c, d], allowing it feasible to tune the local polaron density and precisely pattern the polarons [Supplementary Fig. 10]. It is noted that real crystal defects can also be created and moved in some ionic films[35], which however require much larger voltages, and cannot be annihilated with the tip.

The type-I (up) and type-II polarons can also be manipulated with the tip (Table 1), which however bear some differences to the type-I (down) polarons. First, albeit the type-I (up) polaron can also be moved with both polarity biases, it has a lower hopping barrier. As is seen from Supplementary Fig. 9, the type-I (up) polarons are moved even with the scanning condition of 2.5 V/10 pA, while the type-I (down) polarons in the same field of view are all kept stationary. Second, the bias polarities for creating and erasing type-I (up) polarons are both opposite to those of type-I (down) polarons. And the success rate of manipulating

type-I (up) polarons are also higher. Third, for the type-II polarons, their manipulation conditions more resemble those of the type-I (up) polarons.

The different types of polarons can interestingly be converted in a reversible manner (Supplementary Table). By positioning the tip above a type-I (down) polaron at 3.0 V/4 nA, a current jump occurs, after which it converts to a type-I (up) polaron [Fig. 2e, f]. The reversed transition from type-I (up) polarons to type-I (down) polarons can be realized with − 2.5 V/10 pA [Fig. 2e, f]. With that manipulation protocol, the type-II and type-I (up) polarons are interconverted with the tunneling junction set at 2.8 V/100 pA and −2.5 V/10 pA, respectively [Supplementary Fig. 11].

High-resolution STM imaging determines the occupation site of the polarons. For that, low-bias imaging is required to obtain atomic resolution of $CoCl_2$ simultaneously. At low bias, the type-I (down) and type-I (up) polarons appear as large circular depressions, allowing their centers both best inferred as Co-sites [Supplementary Fig. 12]. Their occupations sites can be more unambiguously determined from their atomic resolution imaging with a decorated tip acquired via controlled dipping onto $CoCl_2$. Three Cl atoms surrounding centers of both the type-I (down) and type-I (up) polarons become bright, forming a trimer geometry [Fig. 3a, b]. The trimers are all oriented along the same direction of the Cl lattice, demonstrating they identically occupy Co-sites. This observation contrasts to another study that determines different occupation sites of the polarons[36]. Atomic resolution of the type-II polaron, imaged with a normal W tip, appear as a single atomic protrusion on Cl [Fig. 3c], indicating it is centered on Cl-site. No lattice distortions are discernable around all types of naturally formed polarons.

## DFT calculated polarons

To reveal the nature of different types of polarons, we perform first-principles calculations with hybrid functional HSE06 on single electron-doped $CoCl_2$ monolayer. The comparison between experimental and simulated STM images are shown in Fig. 3a–c. Type-I (down) and type-I (up) polarons are both associated with a Co-centered state where the polaronic charge localizes on a single Co atom. This localization breaks symmetry in the out-of-plane direction, yielding bowing in either the upward or downward direction, forming type-I (down) and type-I (up) polarons, respectively. Type-II polaron is associated with a Cl-centered state where the polaronic charge is equally distributed around the three nearest neighbor Co atoms. The distribution of polaronic charge and local lattice distortions are shown in Fig. 3d, f, h and Fig. 3e, g, i for Co- and Cl-centered polarons. Note that the atomic relaxation shown in Fig. 3d, with the top layer nearest-neighbor Cl displacing more than those on the bottom layer is for type-I (down) polaron, which is opposite the case of the type-I (up) polaron.

The polaron formation energy is defined as the energy difference between polaronic state and a delocalized state at conduction band edge. Even without lattice distortions, the polaron formation energy is calculated to be −0.03 and −0.05 eV for Co- and Cl-centered polarons, respectively, indicating that electron-electron interactions alone are enough to overcome the kinetic energy increase associated with localization and stabilize the polaron. With atomic relaxation, these energies are further lowered to −0.24 and −0.29 eV for Co- and Cl-centered polarons, respectively. This stability may explain why these polarons are so plentiful in these samples. While the inclusion of a substrate in calculation increases the formation energy of both polarons by about 0.06 eV, the relative energy of the different polarons is unchanged and it has negligible influence on atomic relaxation (<0.01 Å). The density of states in Fig. 3j, k shows that the polaronic states are inside the fundamental gap and the atomic species projection confirm that they are localized to Co atoms.

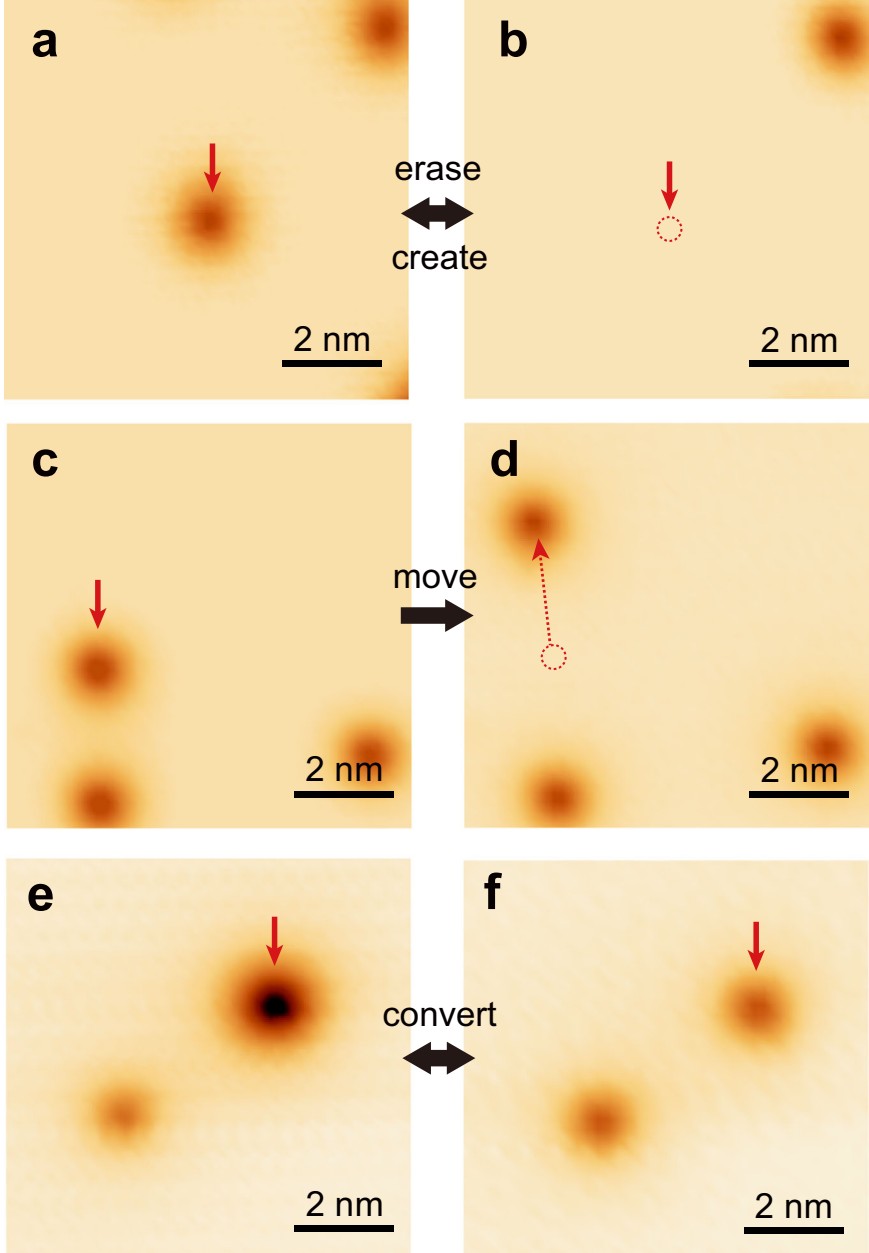

**Fig. 2 | Manipulation and interconversion of polarons. a, b** STM images showing erasing and creating a type-I (down) polaron. **c, d** STM images showing moving a type-I (down) polaron. **e, f** STM images showing interconversion between type-I (down) and type-I (up) polarons. The red dashed circles mark the original location of the polarons before manipulation. The red arrows mark the manipulated polarons.

Experimentally, polaronic state signal is observed in STS only for Cl-centered polaron but our calculations reveal in-gap states for both polarons. We attribute this to the localized charge distribution of polaronic states along the out-of-plane direction, due to the $3d$-orbital nature of these states. As the STM tip is -5 Å higher than the top Cl layer in experiments, much larger than the extent of $3d$-orbitals, the sample-to-tip tunneling is dominated by the wavefunction "tail" of polaronic states, which decays exponentially into the surface region. We find the charge density of Cl-centered polaron is orders of magnitude larger than that of the Co-centered one in the surface region [Supplementary Fig. 13]. As such, in-gap states associated with type-I (down) and type-I (up) polarons (Co-centered) are inherently more difficult to see from STS than the type-II polaron (Cl-centered). Theory also predicts that the type-I (down/up) polarons and type-II polarons would have

significantly different Co-3$p$ core-level shifts of 4.0–4.7 and 1.4–1.8 eV, respectively [Supplementary Fig. 14].

## Manipulation mechanism

Having identified the polaron types and occupation sites, we investigate the physical mechanism behind their manipulations, which are equivalent to controlling the excess electrons. For polaron creation, the excess electron is transferred from the tip or substrate into CoCl$_2$ and gets trapped. Polaron annihilates upon releasing the excess electron back to the tip or substrate. Polaron movement and conversion process are associated with the excess electron hops among equivalent and inequivalent sites, respectively. Our experiments below indicate that the electric field from the STM tip ($E_{tip}$) tunes the energy barrier in all different categories of manipulations, and tunneling

electrons inelastically transfer energy to the excess electron for polaron movement. Despite that clarifying the specific origin for the electric-field-induced barrier change requires another independent study, the $E_{tip}$ may act on the top-Cl anions and the Co cations,

changing their relative height (Fig. 4). As a rule of thumb, reducing their relative height gets the excess electron trapped under the tip, and increasing their relative height releases the excess electron, i.e. causing either polaron annihilation or conversion. Such barrier change requires large-enough $E_{tip}$, unveiling the polaron manipulation requires certain voltages. Because excess electrons of the type-I (down) and type-I (up)[type-II] polarons reside on opposite sides of the Co sites, this also explains why they are manipulated with opposite polarity biases. Moreover, since the polaron has three available electron transfer channels (Fig. 4), either channel can be possibly provoked under the same manipulation, conforming to the observation (Supplementary Table).

### Table 1 | Polaron manipulation conditions

| Manipulation\type | Type-I (down) | Type-I (up) | Type-II |
|---|---|---|---|
| Create | −3.5 V/0.1 nA | 1.5 V/80 nA | 1.4 V/ > 10 nA |
| Move | 3 V/3 nA<br>−3.5 V/0.1 nA | 1.8 V/10 pA<br>− 1.5 V/10 pA | 4 V/10 pA |
| Erase | 3 V/7 nA | −1.7 V/5 nA | − 1.7 V/ 5 nA |

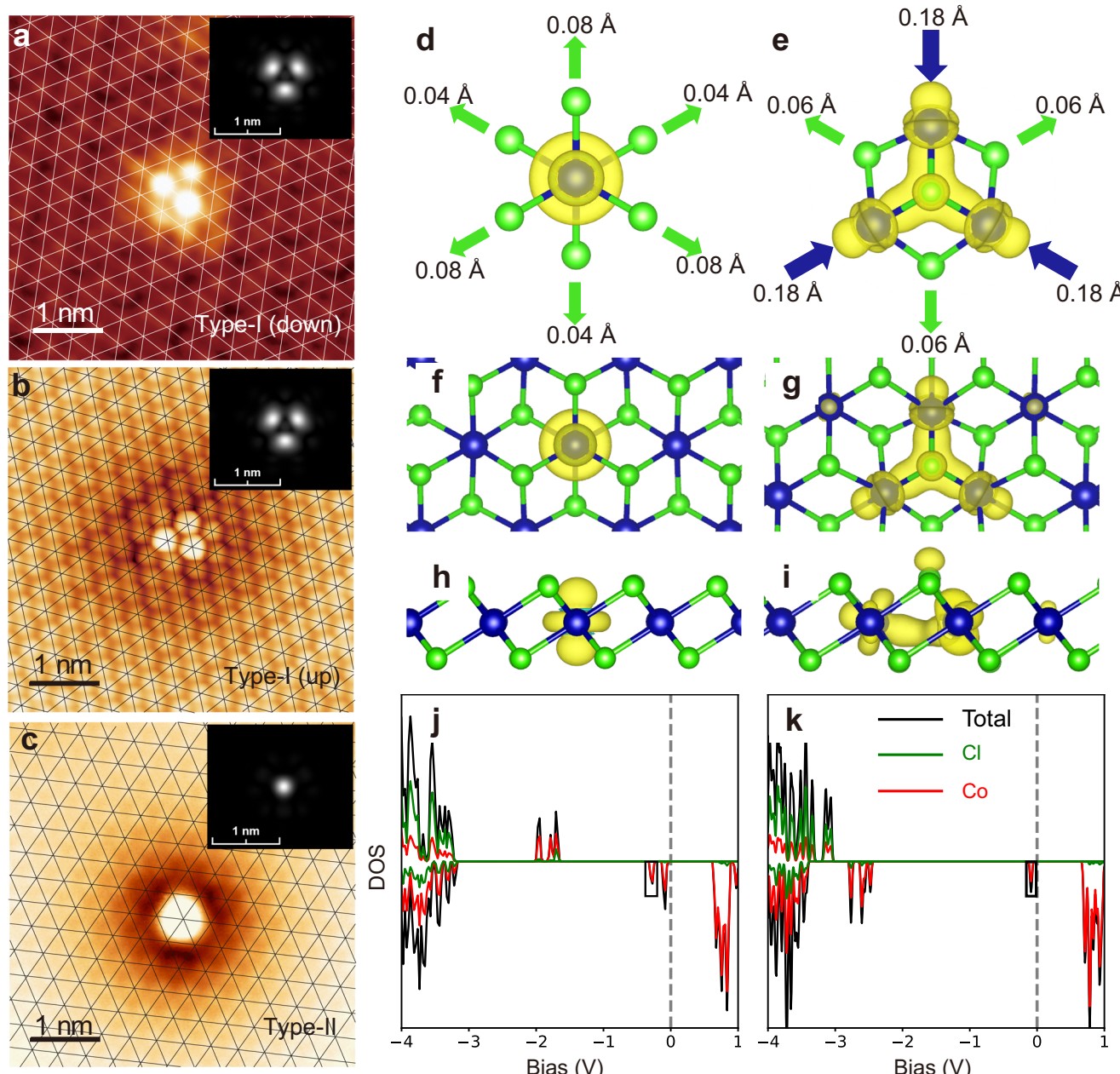

**Fig. 3 | Atomic resolution and calculation of polarons. a** Atomic resolution ($V_s = −1$ V, $I_t = 100$ pA) of type-I (down) polaron. **b** Atomic resolution ($V_s = −0.2$ V, $I_t = 50$ pA) of type-I (up) polaron. **c** Atomic resolution ($V_s = 1.4$ V, $I_t = 10$ pA) of type-II polaron. Insets images are simulated DFT images for corresponding polarons. **d, e** Lattice distortions and polaronic charge densities of (**d**) Co-centered and (**e**) Cl-

centered polarons. Those small lattice distortions are beyond the resolution of STM. **f, g** Top view and (**h, i**) side view of the charge densities of Co-centered and Cl-centered polarons, plotted with an iso-surface of 0.01e/Å³. **j, k** Density of states projected on each species for Co-centered and Cl-centered polarons, with the polaronic state indicated with the black box.

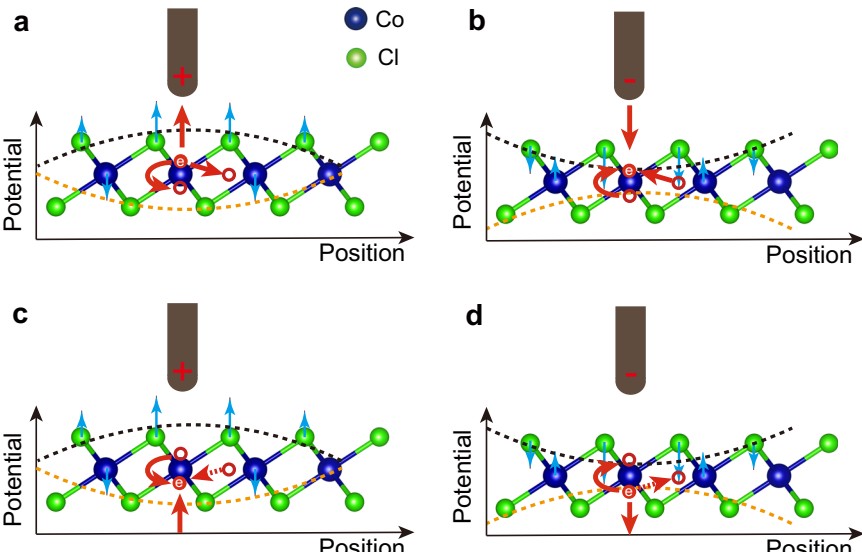

**Fig. 4 | Mechanism of polaron manipulation. a, d** Schematic in (**a**) [(**d**)] showing mechanism of erasure and conversion of type-I (up) [type-I (down)] polarons. For negative sample bias (equivalent to the tip positively biased), top Cl anions move upward and the Co cations move downward, leaving the interlayer distance between Co and top (bottom)-layer Cl stretched (squeezed). This would effectively make the potential for the excess electron on the Co-up (Co-down) site increased (decreased), as depicted with black (orange) dashed line. This results the excess electron on the Co-up (Co-down) site to empty (occupy). The excess electron transfers to the tip (substrate) corresponds to erasure of the type-I (up) [type-I (down)] polaron. The excess electron transfers to the Co-down (triple-Co) site corresponds to conversion from type-I (up) to type-I (down) (type-II) polaron. **b, c** Schematic in (**b**) [(**c**)] showing mechanism of creation and conversion of type-I (up) [type-I (down)] polarons. Note that the red dashed arrows in (**c**, **d**) represent possible polaron conversion routes that are not realized in experiments yet.

Hereafter, we mainly focused on the type-I (up) polaron as an example to justify the above scenario, since its manipulation conditions are mostly controllable. To create polarons, we approached the tip towards $CoCl_2$ surface with a constant *positive* bias and disabled feedback loop. During such a process, the tunneling current increases exponentially until suddenly drops at a critical tip height [Fig. 5a], after which a type-I (up) polaron appears. The current drop comes from the depletion of the LDOS in the presence of the type-I (up) polaron. The critical tip height increases monotonically with increasing bias [Fig. 5b]. Reproducing above measurements using another tip generates a similar dependence but with an evident offset [Fig. 5b]. Since the type-I (up) polaron can only be created with positive bias, this implies the $E_{tip}$ reduces the polaron formation barrier. As such, we modeled the electric field between the tip and substrate, which were considered as a point charge and a semi-infinite metal, respectively, for simplicity [Fig. 5a, inset]. If the polaron formation occurs at a constant $E_{tip}$ value, the following relation applies $V = k(\Delta z + z_0)^2 - \Delta W$, where k is a proportional constant, $Z_0$ ($\Delta z$) represents the initial (decreasing) tip height, and $\Delta W$ is the work function difference between tip and sample [Supplementary Note 3]. This relation fit the experimental data nicely [Fig. 5b], substantiating the above scenario and suggesting the offset for the two tips are caused by their different sharpness. We noticed another study could create polarons with 100% success rate above a low bias of ~0.9 V[36], presumably because a very sharp tip was used in the experiments.

Erasing a polaron also involves a releasing barrier for the excess electron. For that, we similarly approached the tip towards a type-I (up) polaron with disabled feedback loop and a constant *negative* bias. The tunneling current increases until suddenly saturates to the maximum measured value of our preamplifier [Fig. 5c], as caused by the erasure of polaron. The saturated current frequently changes the tip shape, making it difficult to systematically acquire similar data as those of the polaron creation. Nevertheless, the limited number of data points, that were collected without tip change, suggest the critical tip height for polaron annihilation increases with increasing bias magnitude [Fig. 5d]. This substantiates that $E_{tip}$

with an opposite polarity reduces the releasing barrier for the excess electron.

The same operation with a *positive* bias also converts the polaron from type-I (up) to type-I (down) or type-II, namely, two possible converting channels [Fig. 5e]. Reversed polaron conversion from the type-I (down) or type-II back to type-I (up) can be achieved with a *negative* bias [Fig. 5f]. Those observations all consist with the empirical rule of increasing (decreasing) the transfer barrier for the type-I (up) polaron under negative (positive) bias.

Moving polarons are found to be induced by a combined effect of $E_{tip}$ and inelastic electron tunneling (IET). We positioned the tip above a type-I (up) polaron while keeping a constant tunneling current at positive bias with feedback on. This causes change in tip height upon polaron hopping. Typical trace of the change in tip height ($\Delta z$) with time shows two levels of telegraph-like noise [Fig. 6a], due to the polaron hopping between onsite Co and its nearest neighbor (NN) Co sites. This suggests the $E_{tip}$ makes the inward hopping barrier (between NN site and onsite) lower than that of the outward barrier (between NN site and next NN site) [Fig. 6b, inset]. Otherwise, the polaron may hop to next NN site, rendering the two-level telegraph noise stop. The switching rate R increases prominently with increasing bias and tunneling current [Fig. 6b]. The relation between the rate and the bias below 1.7 V fits nicely to an exponential function [Fig. 6b], demonstrating the hopping event is driven by IET process[37]. For each bias, the relation between the switching rate $R$ and the current $I$ fits well to $R \propto I^N$ [Fig. 6c], further substantiating the IET mechanism. The fitting gives $N$ about 1, which demonstrates that the IET event is a one-electron process[38].

## Discussion

In summary, we have visualized and manipulated individual polarons in monolayer $CoCl_2$. Two types of polarons are identified from their distinct signatures of appearances, polaronic states, and occupation sites. Those different types of polarons manifest stunning versatility of manipulation and interconversion. The rich category of polarons envision to exist in other transition metal dihalide films, since they

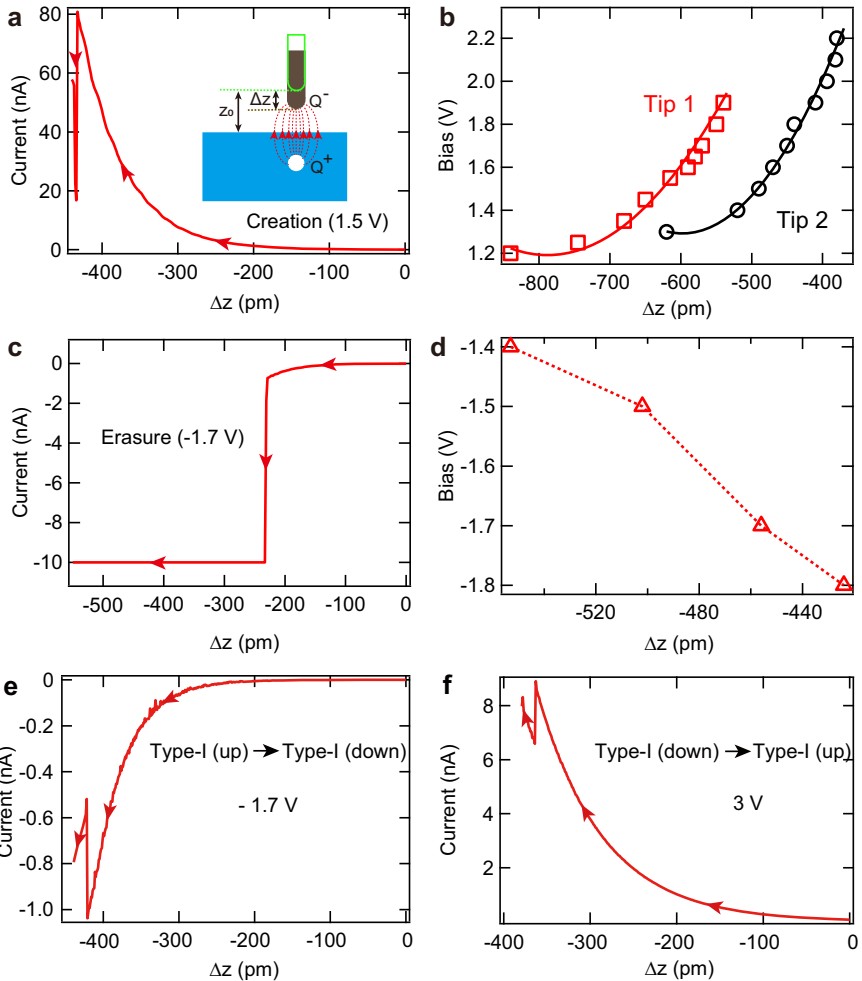

**Fig. 5 | Mechanism of creating, erasing and converting type-I (up) polarons.**
**a** Tunneling current on a clean $CoCl_2$ region with decreasing tip height $\Delta z$ at 1.5 V. The initial tip height $Z_0$ is set at 1.5 V/10 pA. The abrupt current change hallmarks creating a type-I (up) polaron. The inset shows a schematic of the tip electric field. **b** Bias dependent threshold value of $\Delta z$ for creating type-I (up) polarons. The red and black data sets are obtained with two different tips, and red and black curves

are fitting to the respective data. **c** Tunneling current on a type-I (up) polaron as a function of $\Delta z$ at −1.7 V. The abrupt current increase in current hallmarks erasure of the type-I (up) polaron. **d** Bias dependent threshold value of $\Delta z$ for erasing type-I (up) polarons. The dashed line is a guide for the eyes. **e** Tunneling current with $\Delta z$ recorded during conversion from type-I (up) to type-I (down) polaron. **f** Similar as (**e**), but for conversion from type-I (down) to type-I (up) polaron.

share similar electronic and structural characteristics. As such, we have examined monolayer $FeCl_2$ [Supplementary Fig. 15], and indeed observed similar polarons. Moreover, the manipulation of the excess charges trapped inside the polarons render the feasibility of locally tuning the charge density of the films, which acts as a unique knob for controlling other many body states, such as exciton condensation[39]. In addition, the spin properties of the polarons and their exchange interaction with the magnetism of the monolayer insulator constitute another interesting subject of future investigations.

## Methods
### Sample growth and STM measurements
The monolayer $CoCl_2$ film is grown on HOPG. The HOPG substrate was cleaved ex situ and further degassed in a vacuum chamber at approximately 1170 K for 0.5–2 h before growth. High purity ultra-dry $CoCl_2$ powder (99.99% Alfa Aeser) is evaporated at 523 K from a home-made k-cell evaporator and the substrate temperature is kept around 470 K during the sample growth. The base pressure is better than $5 \times 10^{-9}$ Torr. The STM measurement is conducted at 77 K if not stated specifically. An electrochemically etched W wire was used as the STM tip, which had been cleaned on a Ag(111) surface prior to conducting

the measurements. The STS is taken by the lock-in technique with a modulation of 21.2 mV (rms) at 983 Hz.

### First-principles calculations
The VASP code[40] is applied to perform the density functional theory (DFT) calculations with Projector augmented wave (PAW) potentials. We performed hybrid functional (HSE06)[41] calculations to better describe the transition metal system with $3d$ orbitals. The kinetic energy cut-off is 400 eV, and the total energy is converged to $10^{-5}$ eV in self-consistent iterations. The unit cell lattice constant is fully optimized until the atomic forces are less than 0.02 eV/Å. The polarons are simulated by a $5 \times 5 \times 1$ supercell with fixed lattice constant, i.e., only the ions are free to move in the relaxation. To determine the converged polaron configurations was a two-step process. Firstly, we performed PBE + U calculations[42]. The Co-centered polaron forms spontaneously in PBE + U with a standard calculation containing an extra electron. For the Cl-centered polaron, the U parameters were separately chosen for polaron and bulk region in order to localize the polaron to three Co sites centered around Cl. After relaxation was performed, self-consistent PBE + U calculations of the polarons were arrived at without forcing this localization. Using these charge

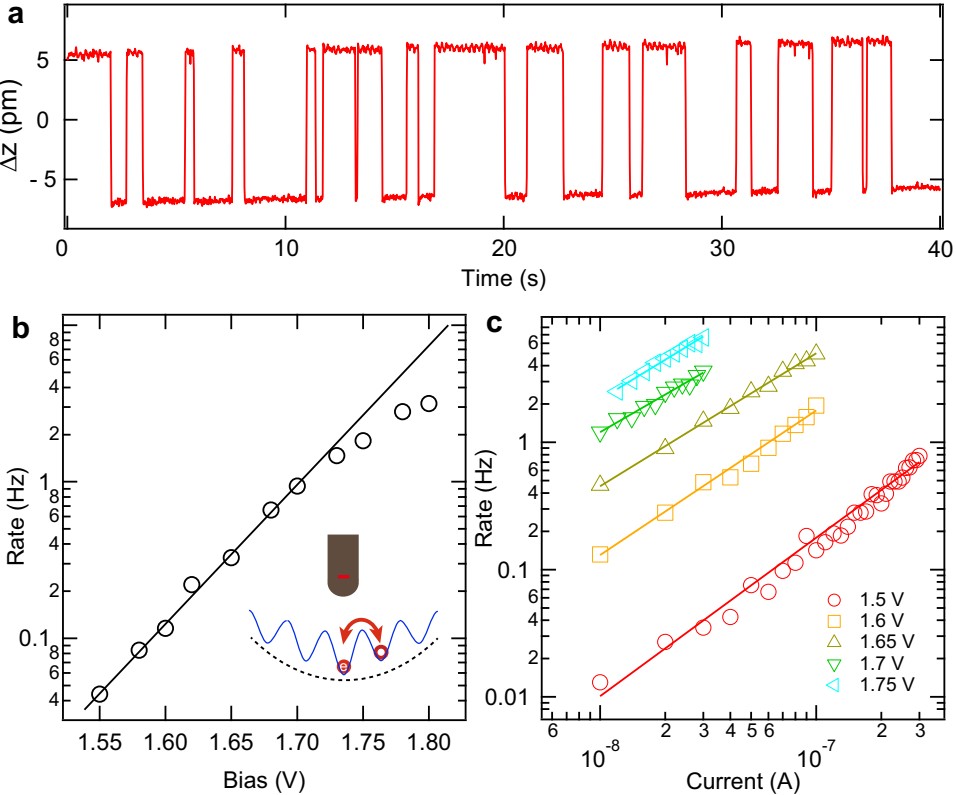

**Fig. 6 | Mechanism of moving type-I (up) polarons. a** Typical trace of tip height switching as a function of time, showing hopping of a type-I (up) polaron between two neighboring Co sites. The trace is recorded at 1.68 V and 10 pA. **b** Bias-dependent switching rate extracted in a similar manner as (**b**). The black line is a fitting to the data below 1.7 V with an exponential relation. Inset is a schematic drawing, showing hopping of the excess electron (red ball and circle) between onsite Co and NN Co sites. The hopping barrier (blue curve) is reduced (dashed curve) by tip electric field. **c** Switching rate of polaron hopping extracted from similar data as (**a**) recorded at different bias and current settings. The solid lines are the fittings to the data with the relation $R \propto I^N$ for different biases. Note that the data in (**b**) and (**c**) are plotted in logarithmic coordinate.

densities and coordinates for the initial electronic structure, both polaron states were eventually relaxed with the HSE06 approximation. To include a substrate requires a $4\sqrt{3} \times 4\sqrt{3}$ moiré supercell with graphene, so that the effect of substrate was studied with PBE + U approximation. We calculated the polaron formation energies (lattice distortions) for both freestanding and substrate-supported supercells with PBE + U approximation and used their difference to estimate the substrate-induced change of polaron formation energies (lattice distortions).

## Data availability
The data that support the findings of this study are available from the corresponding author upon request.

## Code availability
The code that supports the findings of this study is available from the corresponding author upon request.

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

## Acknowledgements

Work in China was funded by the National Key Research and Development Program of China (Grant Nos. 2022YFA1402400, and 2018YFA0307000), the National Natural Science Foundation of China (Grant Nos. U20A6002, 92265201, 11874161, 12047508) and work in the US was funded by the U.S. Department of Energy, Office of Science, Office of Basic Energy Sciences under Award Number DESC-0002623. The supercomputer time sponsored by the National Energy Research Scientific Center (NERSC) under DOE Contract No. DE-AC02-05CH11231 and that by the Extreme Science and Engineering Discovery Environment (XSEDE), under NSF grant number ACI-1548562. SBZ also acknowledges the computational resources from Stampede supercomputer at TACC made available by XSEDE through allocation TG-DMR180114, as well as the Center for Computational Innovations (CCI) at RPI.

## Author contributions

M.C., M.M.P., Z.Y.L. did the experiments with the help of W.H.Z., X.L., L.F.Z..; Y.L., Z.J. did the DFT calculations; Y.S.F., M.C., M.M.P., Y.L., Z.J., D.W., S.Z. analyzed the data. Y.S.F., M.C., Y.L., Z.J. wrote the manuscript with comments from all authors. Y.S.F., S.Z. supervised the project.

## Competing interests

The authors declare no competing interests.
