## [Peer Review File · Nature Communications]

Reviewers' Comments:

Reviewer #1:

Remarks to the Author:

The authors have addressed the comments raised in my original report and the paper can be accepted for publication in present form.

Reviewer #2:

Remarks to the Author:

The authors have addressed all my concerns about their manuscript at a satisfactory manner and greatly improved the manuscript (NCOMMS-23-08861-T). I therefore recommend publication of their manuscript as a Research Article in Nature Communications. One very minor point needs to be addressed/clarified, see below:

(1) In L87-89, p. 4 of the main text, the authors mentioned "Multiple moire patterns are observed upon changing the relative angle between CoCl₂ and graphene layers." I don't think their wording here is totally correct, as they can't actually change the relative angle between the two layers in their system. What they should say should be something like "Multiple moire patterns are observed in surface domains characterized by different relative angles between the grown CoCl₂ layer and the graphene substrate."

No further review is required.

Reviewer #4:

Remarks to the Author:

I have read the author's response and revised manuscript/SM in parallel and comparatively with the latest version of Liu and coworkers titled 'Atomic-scale Manipulation of Single-Polaron in a Two-Dimensional Semiconductor.' The authors have addressed virtually all points raised by the four reviewers and executed new analyses to support their interpretation. The authors have pointed out similarities and differences between their work and Liu's, but did not provide either clear evidence or well-founded arguments distinguishing their interpretation from the one of Liu and coworkers. While some aspects of the presentation need to be improved, as detailed below, my opinion is that this article should not be accepted in Nature Communication, but can be recommended for publication in Scientific Reports with minor modifications.

1) Three out of four reviewers argue that the classification into 4 types of polarons is complicated and, to some extent, confusing.

In their response and revisions, the authors could not convince me that what we see are 4 different types of polarons.

To simplify the reading and allow the reader to capture and appreciate the most important aspects of the paper, the

authors should adopt the following notation. Instead of classifying 4 different types of polarons, the authors could introduce only two types: Co-centered and Cl-centered. The former type-1 and type-2 are Co-centers and differ essentially from the direction (up/dn) of the associated distortion.

The former type-3 and 4 can be classified as Cl-center polarons with different STM appearances (single atomic protrusion vs dark center + 6 NN Cl ions). With this classification, it would be much easier to follow the manipulation process as due to voltage (positive or negative, depending on the specific feature of the Co/Cl polaron) induced hopping events.

If the authors want to keep the present classification they have to provide more convincing quantitatively and qualitatively arguments. In my opinion it is not possible to rigorously define 4 different types of polarons in this context.

2) The authors have extensively discussed the difference between their study and the work of Liu in their response and appreciated their effort and clarity. For the sake of rigor and transparency, they should include a brief statement in the main text and a reference (and remove the 'Note added' at the end of the manuscript).

One of the criticisms raised by the authors of this paper to Liu and coworkers is that the two polarons that Liu and coworkers see are not centered on different atoms (Co and Cl) but rather both centered on Co. Are the authors convinced of this conclusion? From my point of view, the proposed classification would help uniform the outcome of the two papers; in both works the authors see similar polaron types (Co/Cl) that might differ more subtly, possibly due to different sample or measuring conditions.

3) While Table 1 might be useful, I would suggest to move Tab.2 in the SM or even remove it. As stated above, the interesting manipulation is the conversion between Co-centered and Cl-centered polarons, which can be explained and discussed with more clarity. As an example of the confusion mentioned above, in Tab.2 there are two different process 2 -> 1 and 2 -> 3 that are apparently produced by the same conditions, -1.7V/4nA. How can this be? Former type 1 and type 3 become the same only under certain circumstances.

Response Letter to Review

Reviewer #1 (Remarks to the Author):

The authors have addressed the comments raised in my original report and the paper can be accepted for publication in present form.

Thank the reviewer for recommending to publish our paper.

Reviewer #2 (Remarks to the Author):

The authors have addressed all my concerns about their manuscript at a satisfactory manner and greatly improved the manuscript (NCOMMS-23-08861-T). I therefore recommend publication of their manuscript as a Research Article in Nature Communications. One very minor point needs to be addressed/clarified, see below:

(1) In L87-89, p. 4 of the main text, the authors mentioned "Multiple moire patterns are observed upon changing the relative angle between CoCl₂ and graphene layers." I don't think their wording here is totally correct, as they can't actually change the relative angle between the two layers in their system. What they should say should be something like "Multiple moire patterns are observed in surface domains characterized by different relative angles between the grown CoCl₂ layer and the graphene substrate."

No further review is required.

Thank the reviewer for recommending to publish our paper. Following the reviewer's suggestion, we have replaced the sentence with the recommended one

Reviewer #4 (Remarks to the Author):

I have read the author's response and revised manuscript/SM in parallel and comparatively with the latest version of Liu and coworkers titled 'Atomic-scale Manipulation of Single-Polaron in a Two-Dimensional Semiconductor.' The authors have addressed virtually all points raised by the four reviewers and executed new analyses to support their interpretation. The authors have pointed out similarities and differences between their work and Liu's, but did not provide either clear evidence or well-founded arguments distinguishing their interpretation from the one of Liu and coworkers. While some aspects of the presentation need to be improved, as detailed below, my opinion is that this article should not be accepted in Nature Communication, but can be recommended for publication in Scientific Reports with minor modifications.

Thank the reviewer for acknowledging our efforts in addressing all the reviewers' points, the new analyses, and his/her comments that only minor modifications are needed to accept our manuscript.

However, we cannot agree that our paper "did not provide either clear evidence or well-founded arguments distinguishing their interpretation from the one of Liu and coworkers."

We do not have the latest version of Liu's paper that the reviewer is referring to, and could only compare with the original version of arXiv:2205.10731. We have already given detailed descriptions on our different findings compared to the work of Liu et al. in our last reply to review, and would like to briefly summarize the main aspects as following:

1. **Different polaron formation interactions.** We discovered electronic polarons, where electron-electron interaction drives the polaron formation that is further stabilized by electron-phonon interaction, as evidenced from DFT calculations described in Page 8. Liu's paper considered the polarons as conventional electron-phonon polarons.
2. **Different polaron occupation sites.** We totally found three kinds of polarons (two Co-centered and one Cl-centered), while Liu et al. only identified two different states. More important, even though they classified their polarons as Co-centered and Cl-centered, Liu et al. in effect have never observed any Cl-centered polarons experimentally. In contrast, our evidence on the difference between Co- and Cl-centered polarons is from two consistent high resolution STM images in Figs. 3 and S12.
3. **Different polaron creation mechanism.** Our polaron creation involves an energy barrier which is reduced by tip electric field, as evidenced from Figs. 5a,b. However, Liu's work reported the polaron creation as a one-electron process that certainly happens once exceeds 0.9 V (page 15 in Liu's paper). This contradicts with our Figs. 1e,f, S1, S3b, where those STM images were taken at much higher biases than 0.9 V without generating any polarons.
4. **New polaron erasure mechanism.** We found erasing the polaron also involves an energy barrier that is reduced by tip electric field, as is evidenced from Figs. 5 c,d. In contrast, no erasing mechanism was reported in Liu's paper.
5. **New polaron movement mechanism.** We discovered moving polarons is facilitated by a combined effect from tip electric field and inelastic electron tunneling process, as is evidenced from Fig. 6. Liu's paper didn't report the moving mechanism.
6. **Newly extended material system.** We have demonstrated the polaron manipulations could be extended to another system FeCl_2 . Liu's work only focuses on CoCl_2 .

We cannot agree with the reviewer that our paper should not be accepted in Nature Communications, because we believe the novelty of our work meets the high standard and broad readership of the journal. The interesting study of our work has been recognized by all the reviewers, including Reviewer 4, in the last round of review. (Referee 1: "Overall, the work is well done, the text and the discussion are clear and convincing. The topic of polarons is relevant in materials science and condensed matter physics, and the possibility to observe and manipulate them is relevant." Referee 2: "The story itself is interesting, and should be attractive to broad audiences." Referee 3: "the idea of direct imaging of self-trapped electrons is appealing and exciting". Referee 4: "the results reported in this work are interesting".) We are also encouraged that both Reviewers 1 and 2 have recommended to publish our papers in Nature Communications now.

We indeed appreciate the additional issues raised by Reviewer 4, which help to improve our manuscript further. Our point-by-point responses to those issues are below.

- 1) Three out of four reviewers argue that the classification into 4 types of polarons is

complicated and, to some extent, confusing. In their response and revisions, the authors could not convince me that what we see are 4 different types of polarons. To simplify the reading and allow the reader to capture and appreciate the most important aspects of the paper, the authors should adopt the following notation. Instead of classifying 4 different types of polarons, the authors could introduce only two types: Co-centered and Cl-centered. The former type-1 and type-2 are Co-centers and differ essentially from the direction (up/dn) of the associated distortion. The former type-3 and 4 can be classified as Cl-center polarons with different STM appearances (single atomic protrusion vs dark center + 6 NN Cl ions). With this classification, it would be much easier to follow the manipulation process as due to voltage (positive or negative, depending on the specific feature of the Co/Cl polaron) induced hopping events. If the authors want to keep the present classification they have to provide more convincing quantitatively and qualitatively arguments. In my opinion it is not possible to rigorously define 4 different types of polarons in this context.

Thanks for raising this issue. Indeed, type-1 and type-2 polarons are Co-centered, and type-3 and type-4 polarons are Cl-centered. Following this suggestion, we classify two types of polarons according to their occupation sites. The original type-1 and type-2 polarons are newly designated as type-I (down) and type-I (up) polarons, because they are both Co-centered. The original type-3 polarons are newly labelled as type-II polaron, because it is Cl-centered. The original type-4 polaron is not naturally formed, and only appears after manipulation at high temperature, which was thus mentioned only in supplementary information in the last version. To avoid possible complexity, we prefer not to mention it, and delete relevant supplementary note.

With this new category, we note the different types of polarons have the following characteristics of manipulation conditions:

1. While type-I (down) and type-I (up) polarons are both centered on Co, their excess electrons are occupied in Co-down and Co-up configurations, respectively. According to the mechanism shown in Fig. 4, this makes the bias polarity of their manipulating conditions distinctly different.
2. Although type-I (up) and type-II polarons are centered on Co and Cl, respectively, their excess electrons both have up-configuration, and thus share similar manipulation conditions.

2) The authors have extensively discussed the difference between their study and the work of Liu in their response and appreciated their effort and clarity. For the sake of rigor and transparency, they should include a brief statement in the main text and a reference (and remove the 'Note added' at the end of the manuscript).

Following the reviewer's suggestion, we add the following sentence "This observation contrasts to another study that determines different occupation sites of the polarons [36]." to Page 7 and the sentence "We noticed another study could create polarons with 100% success rate above a low bias of ~ 0.9 V [36], presumably because an extremely sharp tip was used in the experiments." to Page 11 of the main text. Liu's paper is cited as Ref. 36.

One of the criticisms raised by the authors of this paper to Liu and coworkers is that the two polarons that Liu and coworkers see are not centered on different atoms (Co and Cl) but rather both centered on Co. Are the authors convinced of this conclusion? From my point of view, the proposed classification would help uniform the outcome of the two papers; in both works the authors see similar polaron types (Co/Cl) that might differ more subtly, possibly due to different sample or measuring conditions.

We are certainly convinced of this conclusion. Liu's paper mentioned that the type-I polarons are more common and bigger in size than type-II polarons [last paragraph in Page 10]. Based on our observation shown in Fig. R1, their ascribed type-I and type-II polarons can be unambiguously assigned to the newly designated type-I (up) and type-I (down) polarons, respectively, which are both Co-centered in our case. However, Liu et al. assigned their type-I as Cl-centered polaron, despite that the STM image and relative population of this polaron (most abundant) is very similar to our type-I (up). In other words, even though they have classified their two polarons as Co-centered and Cl-centered, Liu et al. in effect have never observed any Cl-centered type-II polarons experimentally. In fact, the actual Cl-centered polaron, i.e., the type-II polaron, has a significantly smaller population ratio of $\sim 1:50$ to the type-I (down). Moreover, as has been shown in Fig. S6, the type-II polaron is indiscernible to the type-I (down) polaron at 0.8 V, which is the imaging voltage used in Liu's paper. All those factors may render the type-II polaron easily mixed with the type-I (down) polaron, and thus not recognized by Liu and coworkers.

In our study, we have provided crystal clear evidence from high resolution STM images with both normal and functionalized tips and rigorously supported from theory that the type-I (down) and type-I (up) polarons are both centered on Co, and the newly designated type-II polarons are centered on Cl.

For clarity, we have added a sentence "The population ratio between the type-I (down) and type-II polarons is about 50:1." to Page 5.

Figure R1 STM image taken on CoCl_2 monolayer films that were grown with the substrate kept at 483 K. Image condition: $V_s = 1.2$ V, $I_t = 10$ pA.

3) While Table 1 might be useful, I would suggest to move Tab.2 in the SM or even remove it. As stated above, the interesting manipulation is the conversion between Co-centered and Cl-centered polarons, which can be explained and discussed with more clarity. As an example of the confusion mentioned above, in Tab.2 there are two different process $2 \rightarrow 1$ and $2 \rightarrow 3$ that are apparently produced by the same conditions, $-1.7\text{V}/4\text{nA}$. How can this be? Former type 1 and type 3 become the same only under certain circumstances.

Following the reviewer's suggestion, we have moved Tab. 2 to Page 3 of supplementary information.

We would like to note that the interesting manipulation is not only between the Co-centered and Cl-centered polarons, but also between the Co-up and Co-down polarons. Actually, the latter conversion is more controllable, which helps to unveil the physical mechanism behind such conversions.

Concerning the same conversion condition for the two processes of $2 \rightarrow 1$ and $2 \rightarrow 3$ [which are I (up) \rightarrow I (down) and I (up) \rightarrow II according to the new designation], that is also understandable. As is shown in Figure 4(a) in the main text, when a negative sample bias is applied on a Co-up polaron, there are three available channels for the excess electrons to transfer, inducing different conversion processes. Namely, the electron can transfer from the Co-up site (i.e. type-I (up) polaron) to the STM tip (i.e., polaron erasure), to the Co-down site [i.e. I (up) \rightarrow I (down) conversion] or to the Cl-centered site [I (up) \rightarrow II]. Due to the quantum nature of those three electron transfer channels, the same manipulation condition could possibly provoke any one of the above three conversions.

For clarity, we have added a sentence "Moreover, since the polaron has three available electron transfer channels (Fig. 4), either channel can be possibly provoked under the same manipulation, conforming to the observation (Supplementary Table)." To Page 10.

Reviewers' Comments:

Reviewer #4:

Remarks to the Author:

The authors took again very seriously all concerns I raised in the previous report and addressed them in a rather convincing way. This was appreciated a lot.

The final draft is much more transparent and easier to follow. Moreover, in their response, the authors have clarified the similarities and differences with Lou's article. Therefore, considering that both articles report new and exciting results, which will undoubtedly spur discussion and future research work on this new field (2D polaron), and considering that Nature Communications does not consider the submission date on the archive as the sole reason for declining publication, it is fair to recommend the publication of this article in Nat Comm.

To increase the dissemination of new results and incentive the discussion among interested readers, I would be recommendable to publish both papers in the same NC issue.

REVIEWERS' COMMENTS

Reviewer #4 (Remarks to the Author):

The authors took again very seriously all concerns I raised in the previous report and addressed them in a rather convincing way. This was appreciated a lot.

The final draft is much more transparent and easier to follow. Moreover, in their response, the authors have clarified the similarities and differences with Lou's article. Therefore, considering that both articles report new and exciting results, which will undoubtedly spur discussion and future research work on this new field (2D polaron), and considering that Nature Communications does not consider the submission date on the archive as the sole reason for declining publication, it is fair to recommend the publication of this article in Nat Comm.

To increase the dissemination of new results and incentive the discussion among interested readers, I would be recommendable to publish both papers in the same NC issue.

Thank the reviewer for recommending to publish our paper in Nature Communications.